# Data Decomposition beyond Splitting for Causal Estimation

**Xuelin Yang**[*]
Meta & UC Berkeley
xuelin@berkeley.edu

**Dhruv Singal**[*]
Meta
dsin@meta.com

**Rina Friedberg**
Meta
rinafriedberg@meta.com

**Michael I. Jordan**
UC Berkeley
michael_jordan@berkeley.edu

**Niloy Biswas**
Meta
niloy@meta.com

## Abstract

In modern causal inference, the way we split and utilize data shapes both the efficiency and uncertainty quantification of treatment effect estimates. This manuscript explores emerging data manipulation strategies that go beyond conventional sample splitting. Building on a recent line of work, we introduce data decomposition methods tailored for causal estimation and examine how they can improve the performance of doubly robust estimators. Empirically, we show that these approaches lead to more precise and robust treatment effect estimates.

## 1 Introduction

We study the problem of partitioning data into subsets when estimating the average treatment effect (ATE) in randomized trials and observational studies. This is a common procedure used to prevent double-dipping and avoid selective inference bias [Robins et al., 1994, Scharfstein et al., 1999, Chernozhukov et al., 2018, Lei and Candès, 2021, Guo and Shah, 2024]. Its standard form dates back to Cox [1975] and involves randomly partitioning the dataset by assigning data points into a training set and an inference set. We will refer to this procedure as *data splitting* hereafter.

For causal inference, a key limitation of canonical data splitting is that inference is performed on only a subset of the data points, making it sensitive to outliers or unrepresentative samples [Andrews et al., 2023, Fithian et al., 2014a]. Using smaller subsets inherently reduces the sample size available for each task, which can diminish statistical power. In low signal-to-noise regimes, which are common in many applications such as analysis of experiments in technology companies [Guo et al., 2021, Tay and Wang, 2022, Gu and Wu, 2022, Chou et al., 2025] and economic policy-making [Knaus, 2022], lower statistical power can lead to sub-optimal decision making. In other regimes such as auto-correlated spatial and time series data, it may not even be possible to split the data into independent parts.

Recent work on alternatives of splitting has address similar shortcomings in other settings. They have primarily been developed for general-purpose predictive tasks, where the training subset is typically used for model fitting, selection, or hypothesis generation, while the inference subset is reserved for evaluation, post-selection adjustment, or hypothesis testing, respectively. For example, creating fractional decomposition of individual data points by leveraging distributional properties may help mitigate the impact of outliers in data selection [Tian and Taylor, 2018, Rasines and Young, 2022, Ignatiadis et al., 2023, Neufeld et al., 2024, Leiner et al., 2025]. More efficient use of information

---

[*]Co-first authors.

39th Conference on Neural Information Processing Systems (NeurIPS 2025) Workshop: CauScien: Uncovering Causality in Science Workshop.

has been achieved by exploiting the data left out from data subsets [Fithian et al., 2014b, Hung and Fithian, 2020, Panigrahi, 2023].

These techniques, however, do not directly translate to causal inference, where data splitting is common, but the nature of the tasks across data subsets can be different. Specifically, one subset is often used to estimate nuisance components, such as the propensity score or outcome model, while another is used to estimate the treatment effect itself [Newey and Robins, 2018]. In a similar flavor of data splitting for causal estimation, Athey and Imbens [2016] divide data between samples used to partition a dataset, and those used to estimate heterogeneous treatment effects. Unlike standard prediction problems, the goal of data splitting here is more complex than just minimizing outcome prediction error, as it typically involves obtaining unbiased and efficient estimates of causal quantities, providing accurate uncertainty quantification, and ensuring robustness to model misspecification and confounding biases.

Our manuscript fills this gap by contributing new methodology and analysis tailored to causal inference. We focus on improving data partitioning on doubly robust estimators, which remain consistent provided that either one of the propensity score or the outcome model is correctly specified [Robins et al., 1994, Rotnitzky et al., 1998, Scharfstein et al., 1999, Laan and Robins, 2003, Bang and Robins, 2005, Van Der Laan and Rubin, 2006, Kennedy, 2023]. Among them, we focus on the Augmented Inverse Probability Weighting (AIPW) estimator of Robins et al. [1994], which combines regression-based outcome modeling with inverse probability weighting (IPW) to achieve double robustness. Specifically,

1. We adapt recently proposed data decomposition methods to enhance doubly robust estimators in causal inference. This approach offers practitioners a novel method for conducting causal estimation and opens up new settings for advancing data decomposition techniques.

2. We empirically evaluate decomposition across a range of regimes, demonstrating improved precision in causal effect estimation. This situates decomposition within a broader effort to optimize data usage for causal inference.

## 2 Estimation Framework

### 2.1 Setup: AIPW Estimation for a Partially Linear Regression

We work with the canonical potential outcome framework and identification assumptions [Imbens and Rubin, 2015] [2]. We have a dataset with $n$ data points $\{(X_i, W_i, Y_i(0), Y_i(1))\}_{i \in [n]}$. For each data point $i$, $X_i \in \mathbb{R}^d$ is the $d$-dimensional vector of covariates, $W_i \in \{0, 1\}$ is the binary treatment assignment, $\{Y_i(0), Y_i(1)\}$ are the potential outcomes under control and treated, and $Y_i = Y_i(W_i)$ is the observed outcome. We are interested in estimating the average treatment effect (ATE):

$$\tau^\star = \mathbb{E}[Y_i(1) - Y_i(0)].$$

While the true data generating process for this dataset can be arbitrary, we estimate a Partially Linear Regression (PLR) model [Robinson, 1988], which is specified as

$$W_i \sim \text{Bern}(\pi(X_i)), \quad \text{and} \quad Y_i = f_\theta(X_i) + W_i \tau^\star + \xi_i. \tag{1}$$

Here, $\pi : \mathbb{R}^d \to [0, 1]$ is the propensity score function, $f_\theta : \mathbb{R}^d \to \mathbb{R}$ is a function parameterized by unknown $\theta \in \mathbb{R}^d$, and $\xi_i$ is an error term that is independent of $(X_i, W_i)$. We assume $\xi_i \sim \mathcal{N}(0, \sigma_i^2)$ and further assume $\sigma_i^2$ to be a known.[3]

When constructing estimators for $\pi$ and $f_\theta$, if the imposed functional form or distributional assumptions fail to capture the true data-generating process, it is possible that our model is misspecified and results in a biased ATE estimate. To mitigate its impact, we employ the canonical AIPW estimator.

AIPW is a two-stage estimator, with the first stage performed on the *training set* $\{(X_i^{(\text{t})}, W_i^{(\text{t})}, Y_i^{(\text{t})})\}_{i \in [n^{(\text{t})}]}$ and second stage performed the *inference set* $\{(X_i^{(\text{i})}, W_i^{(\text{i})}, Y_i^{(\text{i})})\}_{i \in [n^{(\text{i})}]}$.

---

[2]We make the standard identification assumptions: SUTVA, unconfoundedness, and strong overlap. Detailed in Section C.

[3]For settings with unknown $\sigma_i^2$, we can extend our setting by using consistent/robust estimators of the population variance.

The training set is used to fit unknown nuisance functions—namely, the outcome regression functions $\widehat{\mu}(w, X)$ for $w \in \{0, 1\}$ and the propensity score $\widehat{\pi}(X)$. Here, $\widehat{\mu}(w, \cdot)$ indicates outcome model given treatment $w \in \{0, 1\}$. If either of these functions is known (e.g., in randomized experiments, the propensity score is known from the treatment assignment mechanism), we simply plug in the known quantity (i.e., let $\widehat{\pi} = \pi$ or $\widehat{\mu} = \mu$, respectively). The inference set is then used to compute the AIPW estimate of the ATE using the estimated (or known) nuisance functions, along with the confidence interval around the estimate:

$$\widehat{\text{ATE}}_{\text{AIPW}} = \frac{1}{n^{(\text{i})}} \sum_{i=1}^{n^{(\text{i})}} \phi_{\text{AIPW}}(X_i^{(\text{i})}, W_i^{(\text{i})}, Y_i^{(\text{i})}) \tag{2}$$

where,

$$\phi_{\text{AIPW}}(X_i^{(\text{i})}, W_i^{(\text{i})}, Y_i^{(\text{i})}) = \frac{W_i^{(\text{i})}}{\widehat{\pi}(X_i^{(\text{i})})} \left( Y_i^{(\text{i})} - \widehat{\mu}(1, X_i^{(\text{i})}) \right) + \widehat{\mu}(1, X_i^{(\text{i})}) \tag{3}$$

$$- \left( \frac{1 - W_i^{(\text{i})}}{1 - \widehat{\pi}(X_i^{(\text{i})})} \left( Y_i^{(\text{i})} - \widehat{\mu}(0, X_i^{(\text{i})}) \right) + \widehat{\mu}(0, X_i^{(\text{i})}) \right).$$

We can further estimate the standard error around the AIPW esimate (and the respective confidence interval) using either a bootstrap or derivations from M-estimation [Lunceford and Davidian, 2004].

## 2.2 Data Decomposition for AIPW

While traditionally AIPW estimation has relied on data splitting to generate the training and inference set, we propose to build upon two recent data partitioning strategies: *data thinning* [Neufeld et al., 2024] and *data fission* [Leiner et al., 2025]. In a nutshell, data fission establishes a framework for splitting a single data point into constituent data points, while data thinning specifically splits a single data point from any convolution-closed distribution into multiple independent additive constituents. The latter framework nests fission mainly for the case of Gaussian/Poisson distributions, while offering additional constructive regimes for more general class of distributions, along with conditional independence of constituents. We refer to both these techniques (along with the family of other strategies which split individual data points) as *data decomposition*.

For our setting of causal inference using AIPW estimation, we adopt the Gaussian and Bernoulli data decomposition constructions. Recall that $\{(X_i^{(\text{t})}, W_i^{(\text{t})}, Y_i^{(\text{t})})\}_{i \in [n^{(\text{t})}]}$ is the training set and $\{(X_i^{(\text{i})}, W_i^{(\text{i})}, Y_i^{(\text{i})})\}_{i \in [n^{(\text{i})}]}$ is the inference set. In data decomposition, $n^{(\text{t})} = n^{(\text{i})} = n$.

**Decomposing the Outcomes**  For each $i$, the covariates are shared between the training and inference sets $X_i^{(\text{t})} = X_i^{(\text{i})} = X_i$. If the outcome model is known, $Y_i^{(\text{t})} = Y_i^{(\text{i})} = Y_i$. Otherwise, we sample *decomposition noise* $Z_i \sim \mathcal{N}(0, \sigma_i^2)$ and construct the decomposed outcomes as

$$Y_i^{(\text{t})} = Y_i + \beta Z_i, \quad \text{, and} \quad Y_i^{(\text{i})} = Y_i - \beta^{-1} Z_i. \tag{4}$$

Here, $\beta$ is the *outcome tuning parameter*, which controls how much of the decomposition noise contaminates the training set: larger $\beta$ corresponds to a noisier training outcome. Note that this also means we sample with the decomposition noise variance $\sigma_i^2$ without loss of generality. When $\beta = 1$, the training and inference outcomes contain equal information, as both have outcome variance $2\sigma_i^2$. As noted by Leiner et al. [2025], this corresponds to a 50/50 split in data splitting when outcome variances are the same across different samples. Lastly, we have $Y_i^{(\text{t})} \perp Y_i^{(\text{i})}$—the training and inference outcomes are conditionally independent.

We discuss implementation details in Appendix A, and decomposing the treatment in Appendix E.

## 3 Empirical Study

We compare data decomposition with data splitting with some commonly studied data generation processes listed in Table 1. In all the cases, we use an Ordinary Least Squares model for AIPW outcome model estimation, and a Logistic Regression model for AIPW propensity model estimation.

| Setting | Outcome Model DGP | Notes |
|---------|-------------------|-------|
| Linear: Homoskedastic | $X_i \sim \mathcal{N}(0, \sigma^2)$ 
 $Y_i \sim \mathcal{N}(X_i^T \beta + \tau^\star W_i, \sigma^2)$ | Simplest setting with sizable signal-to-noise ratio. |
| Linear: Heteroskedastic | $X_i \sim \mathcal{N}(0, \sigma^2)$ 
 $Y_i \sim \mathcal{N}(X_i^T \beta + \tau^\star W_i, \sigma_i^2)$ 
 $\sigma_i^2 \sim \text{BetaPrime}(a, b)$ | Heteroskedasticity and outliers. |
| Non-linear: Athey and Imbens [2016] | $Y_i \sim \mathcal{N}(\eta(X_i) + \frac{2W_i - 1}{2}\kappa(X_i), \sigma^2)$ 
 $\eta(x) = \frac{1}{2}x_1 + \frac{1}{2}x_2 + \sum_{k=3}^{5} x_k$ 
 $\kappa(x) = (x_1 + 1)1\{x_1 > 0\}\tau^\star$ 
 $+ x_2 1\{x_2 > 0\}$ | Non-linear DGP in which covariates only affect the treatment effect when taking positive values. |
| Non-linear: Quadratic | $X_i \sim \mathcal{N}(0, \sigma^2)$ 
 $Y_i \sim \mathcal{N}((X_i^T \beta)^2 + \tau^\star W_i, \sigma^2)$ | Simplest possible non-linear DGP to study misspecified linear outcome AIPW model. |

Table 1: **Data Generating Processes (DGPs).** Details provided in Section A

In Figure 1a for the Linear: Homoskedastic benchmark, while data decomposition and splitting are approximately centered around the true ATE, data decomposition yields a noticeably tighter distribution, indicating reduced variance. This demonstrates that data decomposition not only preserves consistency but also improves statistical efficiency. Notably, this improvement is more pronounced as the dimensionality of the covariates increases—interquartile range for data splitting crosses over the statistical insignificant mark as the dimensionality goes beyond $\sqrt{N}$.

In Figure 1b, we plot the ratio of the confidence intervals (CI of data decomposition over CI of data splitting). While the performance of both methods are similar in low-dimension regimes, the difference becomes more pronounced as we increase the dimensions; data decomposition ATE CIs are <10% of data splitting ATE CIs for all the settings we study in high-dimensional regimes.

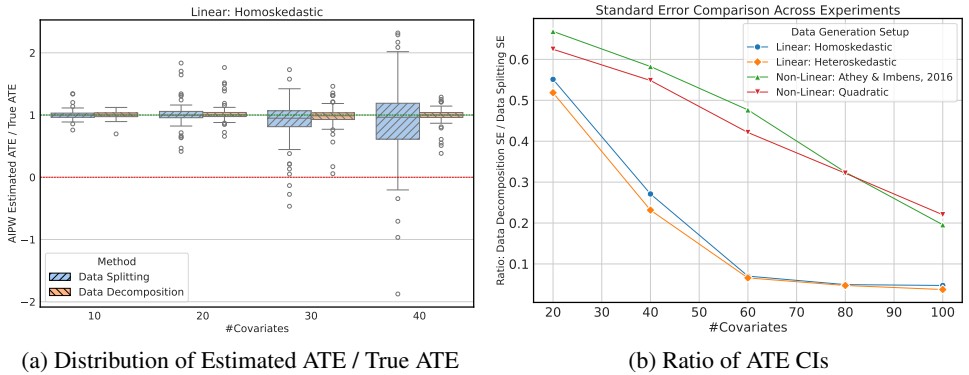

(a) Distribution of Estimated ATE / True ATE       (b) Ratio of ATE CIs

Figure 1: **Comparing data decomposition with data splitting.** In the first panel, we use a linear homoskedastic DGP, and show the distribution of estimated ATEs using AIPW over 100 runs. Results for other DGPs are provided in Figure 2. In the second panel, we use four different DGPs from Table 1, and plot the mean ratio of CIs (Data Decomposition CI / Data Splitting CI) over all runs on increasing number of dimensions.

## 4   Discussion

In this manuscript, we have introduced data decomposition to the AIPW estimator. We demonstrated when this approach is beneficial and under what conditions it may not apply. For future

directions, a deeper investigation into how the treatment assignment mechanism interacts with the data-decomposition strategy would be valuable. Specifically, one may be able to develop adaptive decomposition schemes where the decomposition is informed by observed data characteristics. Second, while we focus on the binary treatment, data decomposition could be adapted to handle multi-valued or continuous treatments, where the structure of splitting may require new theoretical guarantees.

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

# A    Implementation details

**Data splitting**    We use half of the data points as the training set and the other half as the inference set (i.e., $n^{(\text{t})} = \lfloor n/2 \rfloor$, $n^{(\text{i})} = n - n^{(\text{t})}$). This could be easily extended to cases of unequal set sizes (equivalently, of $\beta \neq \beta^{-1}$ in data decomposition).

**Data decomposition**    When decomposing the outcome, we split the information in half by letting $\beta = \beta^{-1} = 1$, corresponding equal splits in data splitting. When decomposing the treatment, we specify $\epsilon$ in the figure captions and conduct an analysis on the effect of $\epsilon$ in Fig. 3 and 4. We discussed the intuition of choosing these parameters in Section 2.2.

**Data Generation**    In Table 1, we use true ATE $\tau^\star = 5$ and generate $n = 1000$ samples. If applicable, we set outcome model noise $\sigma^2 = 1$, whereas in the second case we use $a = 5$ and $b = 3$.

# B    Figures for different DGPs

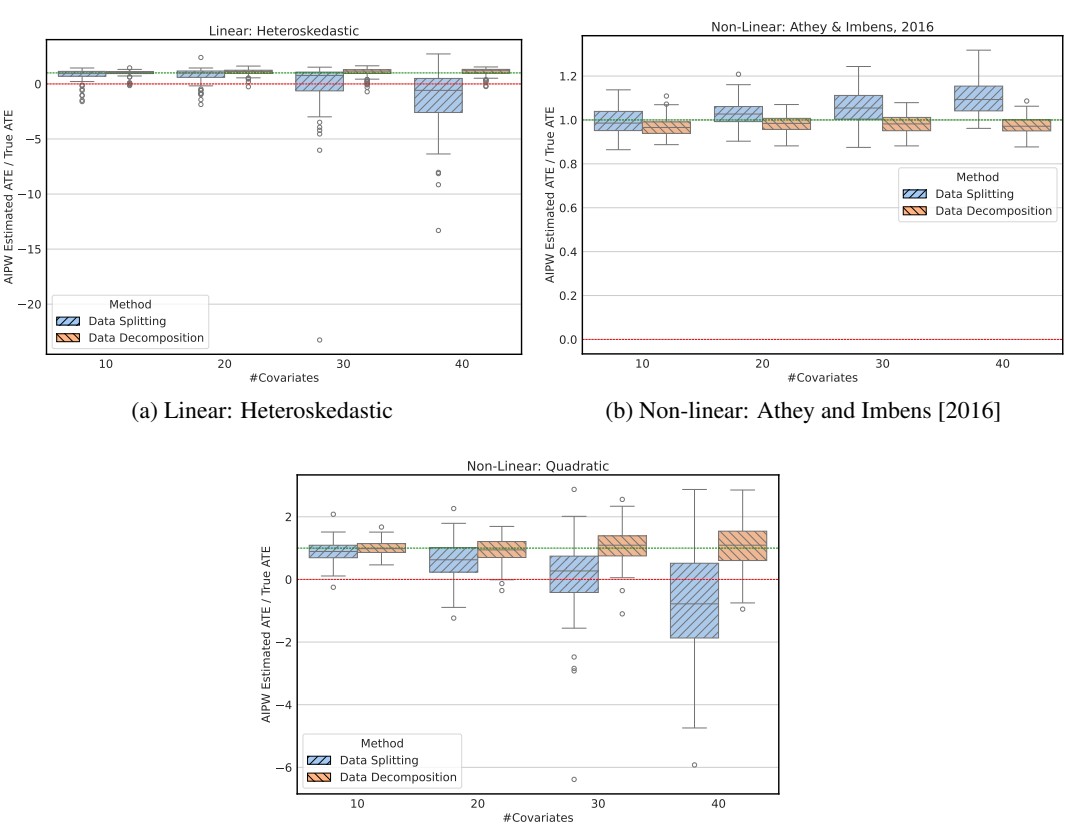

(a) Linear: Heteroskedastic                    (b) Non-linear: Athey and Imbens [2016]

(c) Non-linear: Quadratic

Figure 2: Different DGPs from Table 1.

# C    Identification assumptions

We assume unconfoundedness (*i.e.*, $\{Y_i(0), Y_i(1)\} \perp W_i \mid Y$), bounded second moment for potential outcome $\mathbb{E}[Y_i^2(w)] < \infty$, and strong overlap (*i.e.*, $\eta \leq \pi(x) \leq 1 - \pi(x)$ where $\pi(x) = \mathbb{P}(W_i = 1 \mid X_i = x)$ is the propensity score).

## D  Additional analysis

We now provide further interpretations of the data decomposition method, including the choice of $\beta$, construction of confidence intervals for the outcome model via Fisher information, and connections to prior work.

**Decomposing the outcome with $\beta = 1$**  When fitting the outcome model, the outcome $Y_i^{(t)}$ is obtained from adding the noise $Z_i$, which can be viewed as doubling the variance of the noise when $\beta = 1$. When $n \gg d$, the error of outcome model treatment coefficient (i.e., $\mu(1, X) - \mu(0, X)$) scales approximately with $\sqrt{2}\sigma/\sqrt{n}$, which is at the same order as $\sigma/\sqrt{n/2}$ for data splitting.

**Confidence interval for the fitted outcome model.**  For a MLE $\widehat{\theta}$ with i.i.d. data $W^n = (W_1, \ldots, W_n)$. We denote $\theta^\star$ as the true parameter and $I_W(\cdot)$ as the Fisher information at a single data point. Then, as $n \to \infty$,

$$\sqrt{n}(\widehat{\theta} - \theta^\star) \xrightarrow{d} \mathcal{N}(0, I_W^{-1}(\theta^\star)),$$

or, equivalently,

$$(\widehat{\theta} - \theta^\star) \stackrel{d}{\approx} \mathcal{N}(0, 1/(nI_W(\theta^\star))).$$

This would give a 95% CI

$$(\theta^\star - 1.96\sqrt{n^{-1}I_W^{-1}(\theta^\star)}, \theta + 1.96\sqrt{n^{-1}I_W^{-1}(\theta^\star)}).$$

By Rasines and Young [2022] and Leiner et al. [2025], denoting $S$ as a possible way to choose a training set,

$$I_{\{Y_i^{(t)}\}_{i=n}}^{-1}(\theta) \leq \mathbb{E}[I_S^{-1}(\theta)]$$

where $Y_i^{(t)}$ is the decomposed $Y_i$. Here, the expectation is taken with respect to possible ways of splitting.

**Treatment decomposition: connection to Leiner et al. [2025], Neufeld et al. [2025]**  Neufeld et al. [2025] proposed an improved method for P2 fission (where training set and inference set not independent) of Leiner et al. [2025] in logistic regression using offset adjustment. In their setting, logistic regression was fitted on the inference set after covariate selection on the training set. To account for the data split, they introduced an offset to the logistic regression model: specifically, $\log(\epsilon/(1-\epsilon))$ when $A_i^{(t)} = 0$, and $\log((1-\epsilon)/\epsilon)$ when $A_i^{(t)} = 1$. However, in the case of the AIPW estimator, the propensity score model is not re-fitted during the inference stage. Therefore, unlike the approach in the discussion paper, we do not modify the logistic regression model itself; instead, we directly adjust the propensity scores on the inference set using the known data thinning mechanism.

**Connection to data carving [Fithian et al., 2014b]**  Data carving operates by conditioning on the selected model in the first stage and reuse the information left out, meanwhile maintaining the type I error control. In comparison, data fission performs an a priori partition of the data, allocating separate portions for training and inference, therefore, similar to splitting, it preserves independence without requiring post-selection adjustments (i.e., satisfies Equation 4 in the Fithian et al. [2014b]).

## E  Decomposing the Treatments

If the propensity score model is known, $A_i^{(t)} = A_i^{(i)} = A_i$. Otherwise, we follow Neufeld et al. [2025] and sample *treatment noise* $Q_i \sim \text{Bern}(\epsilon)$ and construct the decomposed treatments as

$$A_i^{(t)} = (1 - Q_i)A_i + Q_i(1 - A_i), \quad \text{and} \quad A_i^{(i)} = A_i. \tag{5}$$

Here, $\epsilon \in [0, 1]$ is the *treatment tuning parameter*; when $Q_i = 1$, we flip the original sample as the training sample. This is akin to adding uncertainty *proportional to* $\epsilon$ into the observed propensity score. As $\epsilon \to 0$, the training treatment concentrates on the observed $A_i$, recovering the original

assignment. As $\epsilon \to 0.5$, the training treatment becomes nearly independent of the true assignment, making the model essentially oblivious to the true treatment.

When the true propensity score is unknown and the distribution of treatments in the training set is shifted via data decomposition, it is necessary to adjust for this shift during inference. Instead of using the standard propensity score $\mathbb{P}(A_i^{(\mathrm{i})} = 1 | X_i)$, we utilize the posterior distribution $\mathbb{P}(A_i^{(\mathrm{i})} = 1 | A_i^{(\mathrm{t})}, X_i)$, which accounts for the observed treatment in the training set and the covariates.

We compute the propensity scores on inference set as follows for Eq. 3:

$$\widehat{\pi}(X_i) = \widehat{\mathbb{P}}(A_i^{(\mathrm{i})} = 1 | A_i^{(\mathrm{t})}, X_i) = \frac{\widehat{\mathbb{P}}(A_i^{(\mathrm{t})} = 1 | X_i)}{\widehat{\mathbb{P}}(A_i^{(\mathrm{t})} = 1 | X_i) + \widehat{\mathbb{P}}(A_i^{(\mathrm{t})} = 0 | X_i)\left(\epsilon/(1 - \epsilon)\right)^{2A_i^{(\mathrm{t})} - 1}}.$$

While this follows a similar approach in Section 3.2 of Neufeld et al. [2025], our setting differs to Leiner et al. [2025] (where no adjustment at the inference stage) and Neufeld et al. [2025] (where the logistic model offset is adjusted at the inference stage). We use the AIPW estimator, where the propensity score model is not re-estimated on the inference set. As a result, rather than modifying a fitted logistic regression model à la Neufeld et al. [2025], we directly adjust the propensity scores themselves on the inference set to reflect the treatment assignment mechanism conditioning on the training set. To our knowledge, this form of adjustment is novel, and is suited to the context of doubly robust estimation under sample splitting.

### E.1 Parameter $\epsilon$ for decomposing treatment in unknown propensity score case

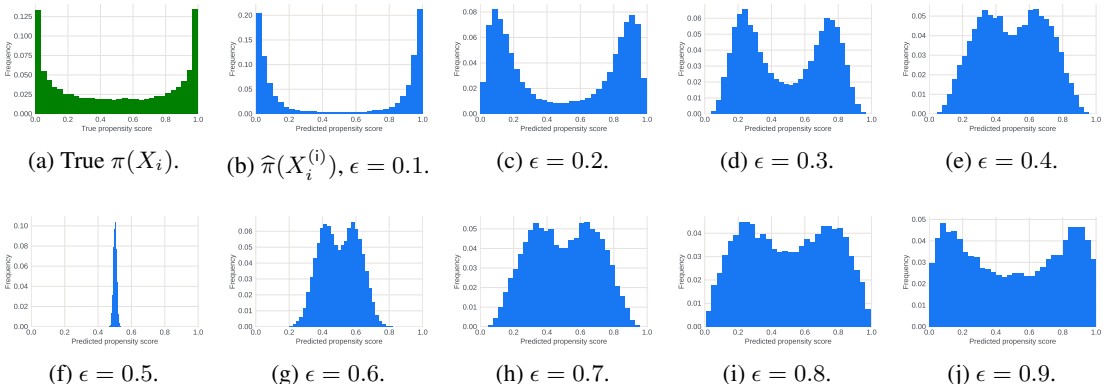

(a) True $\pi(X_i)$.  (b) $\widehat{\pi}(X_i^{(\mathrm{i})})$, $\epsilon = 0.1$.  (c) $\epsilon = 0.2$.  (d) $\epsilon = 0.3$.  (e) $\epsilon = 0.4$.

(f) $\epsilon = 0.5$.  (g) $\epsilon = 0.6$.  (h) $\epsilon = 0.7$.  (i) $\epsilon = 0.8$.  (j) $\epsilon = 0.9$.

Figure 3: Impact of the flipping probability $\epsilon$ for treatments. $n = 20000, \eta \sim \mathcal{N}(0, \mathbb{I})$. Fig. 3a: distribution of true propensity scores generated by sigmoid($\eta^\top X_i$). Fig. 3b-3j: distribution of data decomposition's propensity scores under different $\epsilon$. Since $A_i \sim \mathrm{Bern}(\mathrm{sigmoid}(\eta^\top X_i))$, when $\eta^\top X_i$ has a large magnitude, the resulting propensity scores become extreme. This leads to unstable estimates for methods based on data splitting, caused by sensitivity to those data points receiving very large weights in AIPW. Instead, data decomposition *smoothens* the propensity score distribution, yielding narrower CIs

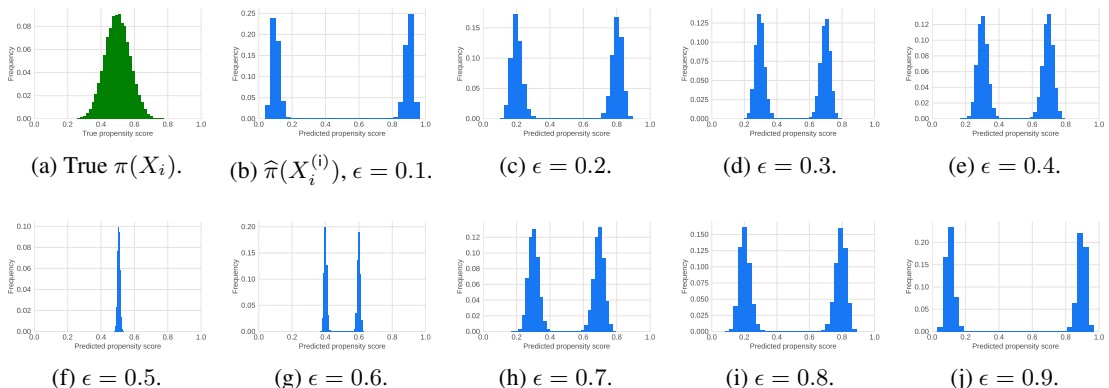

(a) True $\pi(X_i)$.    (b) $\widehat{\pi}(X_i^{(i)})$, $\epsilon = 0.1$.    (c) $\epsilon = 0.2$.    (d) $\epsilon = 0.3$.    (e) $\epsilon = 0.4$.

(f) $\epsilon = 0.5$.    (g) $\epsilon = 0.6$.    (h) $\epsilon = 0.7$.    (i) $\epsilon = 0.8$.    (j) $\epsilon = 0.9$.

Figure 4: Impact of the flipping probability $\epsilon$ for treatments. $n = 20000, \eta \sim \mathcal{N}(0, 0.01 \cdot \mathbb{I})$. Fig. 4a: distribution of true propensity scores generated by sigmoid($\eta^\top X_i$). Fig. 4b-4j: distribution of data decomposition's propensity scores under different $\epsilon$.

