# OpenReview forum: "Data Decomposition beyond Splitting for Causal Estimation"
_NeurIPS.cc/2025/Workshop/Reliable_ML — NeurIPS 2025 - Reliable ML Workshop_

### Official Review · Reviewer_2K3Y · 2025-09-10
**A promising methodological advance in causal inference via data decomposition**

**Rating:** 7
**Confidence:** 4

**Review:**

This paper introduces data decomposition as an alternative to the conventional data splitting paradigm in causal inference, focusing on improving the performance of doubly robust estimators such as AIPW. The authors argue that splitting data into disjoint training and inference sets reduces effective sample size and statistical power, and they propose decomposition strategies (drawing from data thinning and data fission) that allow each data point to contribute to both stages while preserving conditional independence.

Strengths:
* The motivation is clear and compelling: standard data splitting is wasteful and often problematic in high-dimensional or noisy regimes, and the proposed approach addresses this limitation.
* The methodological contribution is novel in the causal inference literature. The authors carefully adapt decomposition strategies originally designed for prediction tasks and provide tailored formulations for both outcomes and treatments in the AIPW framework.
* The empirical results are convincing. Across a variety of data generating processes, decomposition consistently yields narrower confidence intervals and reduced variance relative to splitting. The improvements are especially notable in high-dimensional settings, which are of practical importance.
* The paper is well-structured, situating the work clearly within the literature and highlighting both connections to and distinctions from related approaches (cross-fitting, data carving, selective inference).

Weaknesses / Limitations:
* While the empirical evaluation is extensive, the benchmarks are somewhat stylized (linear, quadratic, and synthetic nonlinear setups). It would strengthen the paper to include results on semi-synthetic or real-world datasets where decomposition might demonstrate practical advantages or reveal limitations.
* The discussion of failure cases is somewhat brief. For instance, the paper notes that cross-fitting may outperform decomposition when outcome models are severely misspecified, but the analysis of why this occurs could be deepened.
* The treatment decomposition component, while interesting, is more complex and may be harder for practitioners to apply correctly. The paper could benefit from additional guidance or heuristics for choosing tuning parameters such as β and ϵ in practice.
* Some connections to existing work (e.g., recent advances in targeted maximum likelihood estimation or Bayesian causal inference frameworks) could be more thoroughly explored to position the contribution within the broader ecosystem.

---

### Official Review · Reviewer_yY7K · 2025-09-14
**Data Decomposition beyond Splitting for Causal Estimation**

**Rating:** 8
**Confidence:** 3

**Review:**

# Summary
Authors build upon several recent data decomposition works in the specific setting of Augmented Inverse Probability Weighting (AIPW) estimation. They describe a couple of methodologies for data decomposition in this setting and present empirical results of these methods in 4 different data generation regimes as compared to standard data splitting. Empirical results indicate noticeable improvements in variance and confidence interval width particularly as dimensionality increases.

# Strengths
1. I appreciate the clear interpretation of figures on lines 112-121, empirical results are clearly described and present compelling improvements over the standard splitting methods
2. Decomposition methodology is well-stated and understandable with clear references to related work
3. Paper is overall well-written and the contribution is distinctly stated

# Weaknesses/Limitations
1. This is outside the scope of this primarily empirical workshop paper, but it would be interesting to see some more explicitly discussed theoretical guarantees, if any exist.

# Suggestions for Authors
There's a typo in line 23, "has address" should be either "address" or "has addressed".

There are no glaring issues with this work in the context of the 4 page workshop track, as stated above, the empirical results are very compelling and clearly support the authors' goal of presenting improved data decomposition methods for AIPW. For later versions, It might be useful to make brief mention of what empirical results show (i.e. variance reduction, CI width decrease, etc) in the introduction to prepare the reader for these results. If possible, it would also be interesting to see any sort of results on real datasets. As data decomposition methods are novel techniques for practitioners, comparison to AIPW with standard splitting on a commonly used dataset could be a compelling addition to the synthetic versions.